# Atmospheric aerosol compositions and sources at two national background sites in northern and southern China

Qiao Zhu[1], Ling-Yan He[1], Xiao-Feng Huang[1], Li-Ming Cao[1], Zhao-Heng Gong[1,*], Chuan Wang[1], Xin Zhuang[1], Min Hu[2]

[1]Key Laboratory for Urban Habitat Environmental Science and Technology, School of Environment and Energy, Peking University Shenzhen Graduate School, Shenzhen, 518055, China.

[2]State Key Joint Laboratory of Environmental Simulation and Pollution Control, College of Environmental Sciences and Engineering, Peking University, Beijing, 100871, China.

[*] Now at: John A. Paulson School of Engineering and Applied Sciences, Harvard University, Cambridge, Massachusetts, 02138, USA.

**Abstract.** Although China's severe air pollution has become a focus in the field of atmospheric chemistry and the mechanisms of urban air pollution there have been researched extensively, few field sampling campaigns have been conducted at remote background sites in China, where air pollution characteristics on a larger scale are highlighted. In this study, an Aerodyne high-resolution time-of-flight aerosol mass spectrometer (HR–ToF–AMS), together with an aethalometer, was deployed at two of China's national background sites in northern (Lake Hongze site, 33.23 °N, 118.33 °E, alti. 21 m) and southern (Mount Wuzhi site, 18.84 °N, 109.49 °E, alti. 958 m) China in the spring seasons in 2011 and 2015, respectively, in order to characterize submicron aerosol composition and sources. The campaign-average $PM_1$ concentration was $36.8 \pm 19.8$ μg m$^{-3}$ at the northern China background (NCB) site, which was far higher than that at the southern China background (SCB) site ($10.9 \pm 7.8$ μg m$^{-3}$). Organic aerosol (OA) (27.2%), nitrate (26.7%), and sulfate (22.0%) contributed the most to the $PM_1$ mass at NCB, while OA (43.5%) and sulfate (30.5%) were the most abundant components of the $PM_1$ mass at SCB, where nitrate only constituted a small fraction (4.7%) and might have contained a significant amount of organic nitrates (5–11%). The aerosol size distributions and organic aerosol elemental compositions all indicated very aged aerosol particles at both sites. The OA at SCB was more oxidized with a higher average oxygen to carbon (O/C) ratio (0.98) than that at NCB (0.67). Positive matrix factorization analysis (PMF) was used to classify OA into three components, including a hydrocarbon-like component (HOA, attributed to fossil fuel combustion) and two oxygenated components (OOA1 and OOA2, attributed to secondary organic aerosols from different source areas) at NCB. PMF analysis at SCB identified a semi-volatile oxygenated component (SV-OOA) and a low-volatile oxygenated component (LV-OOA), both of which were found to be secondary species and could be formed from precursors co-emitted with BC. Using the total potential source contribution function model, the likely source areas of the major $PM_1$ components at both sites were a large

---

*Correspondence to*: X.-F. Huang (huangxf@pku.edu.cn)

regional scale in East Asia. The possible sources may not only include emissions from the Chinese mainland but also include emissions from ocean-going cargo ships and biomass burning in neighboring countries.

## 1. Introduction

With the rapid economic growth and urbanization, severe events of poor air quality, characterized by high concentrations of
fine particles ($PM_{2.5}$), frequently occurred in China. Average $PM_{2.5}$ concentrations across China have been documented, with high levels appearing in northern regions and relatively low levels in southern areas (van Donkelaar et al., 2010; Yang et al., 2011). Many measurements and source analyses of ambient aerosols were conducted in different areas of China, giving local aerosol composition and sources. For example, Huang et al. (2014) found that, during a single winter pollution event, the levels, compositions, and sources of $PM_{2.5}$ are significantly different for four cities in China. The summary of $PM_{2.5}$
measurement based on off-line filter sampling across China by Yang et al. (2011) also revealed largely variable compositions for different areas. Since 2006, valuable insights on the composition, sources, and evolution processes of submicron particles in China were obtained through a dozen of field campaigns using various types of some powerful online tool. Aerodyne aerosol mass spectrometer (AMS) instruments, capable of on-line measuring chemical composition of non-refractory submicron aerosol species (Canagaratna et al., 2007; Ng et al., 2011b).These previous campaigns mostly focused
on much polluted areas in eastern China, such as the Beijing–Tianjin–Hebei area (Takegawa et al., 2009; Huang et al.,2010; Sun et al., 2010, 2012, 2013, 2015; Zhang et al., 2011; Hu et al., 2013), the Yangtze River Delta (Huang et al., 2012, 2013), and the Pearl River Delta (He et al.,2011; Xiao et al., 2011; Lee et al., 2013), Xu et al. (2014) also reported AMS measurement results in a western city of Lanzhou in China. In addition, Wang et al. (2016) recently used an Aerodyne soot particle-aerosol mass spectrometer (SP-AMS), for the first time in China, to investigate the occurrence of fullerene soot in
ambient air. According to these studies, organic aerosol (OA) was commonly found to be the primary aerosol component, accounting for more than thirty percent of the total measured particle mass (e.g., Takegawa et al., 2009; Huang et al., 2010,2011,2012; Sun et al., 2015). Furthermore, positive matrix factorization (PMF) analysis based on the OA mass spectra was used to separate OA into several factors that indicate different aerosol sources. Specifically, a hydrocarbon-like OA (HOA) factor, attributed to primary emissions associated with oil combustion, and an oxygenated OA (OOA) factor,
attributed to photo-chemically formed secondary organic matter, were most frequently distinguished and quantified, while cooking OA, biomass burning OA, and coal burning OA were also identified in some field campaigns (e.g., Huang et al., 2010, 2011, 2012; He et al., 2011; Hu et al., 2013). Recently, a novel PMF procedure, using the multi-linear engine (ME-2), was developed to apportion the OA sources in Beijing and Xi'an of China, allowing for a better selection of the source apportionment solution (Elser et al., 2016). However, nearly all previous AMS studies in China focused on sources and
chemical properties of aerosol particles in the very polluted urban or urban downwind areas with strong local source emissions, and thus these results cannot well reflect the general air pollution characteristics in a larger regional scale. The regional air pollution has been found to not only determine air quality in rural or remote areas, but also be a critical factor in

determining urban air quality in China, such as in Beijing, Shanghai, and Shenzhen (Huang et al., 2010; He et al., 2011; Huang et al., 2012).

So far, several measurements and source analyses based on AMS have been conducted at background sites around the world. Sun et al. (2009) reported the composition and size distribution of NR-PM$_1$ at the Whistler Peak in Canada. Chen et al. (2009, 2015) conducted an AMS study to characterize submicron biogenic organic particles in the Amazon Basin. Ovadnevaite et al. (2011) demonstrated the occurrence of primary marine organic aerosol plumes on the west coast of Ireland. Du et al. (2015) described the aerosol composition using an ACSM at a national background monitoring station in the Tibetan Plateau in western China. These background site aerosol studies were all conducted in remote areas, which represent global background atmosphere rather than regional background atmosphere. However, regional background atmosphere is more critical to reflect the general picture of anthropogenic emissions in hot polluted regions. In this study, we performed online aerosol measurement field campaigns at two national air background sites in both northern and southern region in eastern China, which has a population of more than one billion and is characterized by high air pollution levels, as a result of high urbanization and industrialization. An Aerodyne high-resolution time-of-flight aerosol mass spectrometer (HR–ToF–AMS), together with an aethalometer, was deployed at the two background sites to characterize and compare submicron aerosol compositions and evolution processes. The AMS datasets were then further analyzed based on the positive matrix factorization (PMF) method and the total potential source contribution function (TPSCF) model to get more insights on particle source categories and origins in eastern China.

## 2. Experimental methods

### 2.1. Sampling sites and meteorological conditions

The data presented in this study were collected at the Lake Hongze site from March to April in 2011 and at the Mount Wuzhi site from March to April in 2015. The Lake Hongze site (33.23 °N, 118.33 °E, alti. 21 m) is located at the center of a wetland natural reserve in Jiangsu province in northern China. Jiangsu, as one of the most affluent areas of China, has a dense population and is highly industrialized. The Mount Wuzhi site (18.84 °N, 109.49 °E, alti. 958 m) is located on the highest peak of a mountain on Hainan province in southern China. Hainan is separated from the mainland China by a narrow strait and is famous for tourism due to its beautiful natural scenery. Figure 1 shows the locations of both sites. These two sampling sites are relatively far from urban and industrial areas, serving as official national air quality background sites of China. Spring time was chosen as the intensive observation period because the wind in this season is the most variable in the whole year, and thus, more representative particles for each site could be collected. Although the sampling years were different for the two sites, the year-to-year difference of air pollution at one site was far smaller than the difference between northern China and southern China. According to the official monitoring data (http://www.jshb.gov.cn/), the annual average PM$_{2.5}$ mass concentration in Jiangsu Province was 73 µg m$^{-3}$ for 2013, 66 µg m$^{-3}$ for 2014, and 57 µg m$^{-3}$ for 2015, while that in Hainan Province was only 26 µg m$^{-3}$ for 2013, 28 µg m$^{-3}$ for 2014, and 20 µg m$^{-3}$ for 2015

(http://www.ep.hainan.gov.cn/). In addition, the high altitude of the Mount Wuzhi site would also not make a big difference of air pollution from the ground site in Hainan Province due to its weak anthropogenic emissions. During the sampling period in 2015, the average $PM_{2.5}$ mass concentration at the Mount Wuzhi site was 15 μg m$^{-3}$ and that at another ground site on the foot of the mount, about 60 km to the south, was quite similar (16 μg m$^{-3}$). The meteorological data obtained during the two campaigns are summarized in Table 1, and the corresponding near-ground air mass back-trajectories are plotted in Figure 1.

**Table 1.** Meteorological conditions and $PM_1$ species concentrations during NCB and SCB campaigns.

| Sampling site | | NCB | SCB |
|---|---|---|---|
| Sampling period | | Mar 19 to Apr 24, 2011 | Mar 18 to Apr 15, 2015 |
| Meteorology | $T$ (˚C) | 13.2±5.0 | 22.6±3.6 |
| | RH (%) | 56.9±10.6 | 87.4±16.3 |
| | WS (m s$^{-1}$) | 4.0±2.6 | 3.6±2.5 |
| Species (μg m$^{-3}$) | Org | 9.8±5.6 | 4.9±3.6 |
| | $SO_4^{2-}$ | 7.7±5.0 | 3.4±2.8 |
| | $NO_3^-$ | 9.4±6.7 | 0.5±0.6 |
| | $NH_4^+$ | 5.5±3.2 | 1.5±1.1 |
| | Chl | 0.3±0.3 | 0.03±0.03 |
| | BC | 2.6±1.7 | 0.7±0.5 |
| | $PM_1$ | 36.8±19.8 | 10.9±7.8 |

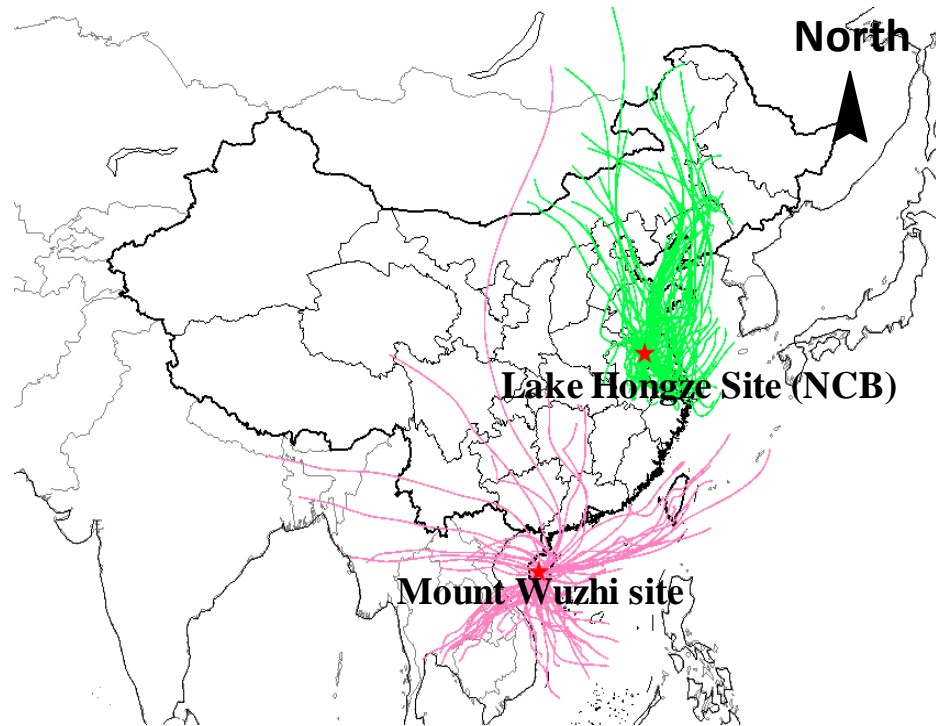

**Figure 1.** Locations of the Lake Hongze site (NCB) and Mount Wuzhi site (SCB). The green and pink lines represent the typical 48 h near-ground (100 m above ground level) air mass back-trajectories during the sampling campaigns.

## 2.2. HR–ToF–AMS operation

5    As the details regarding the instrumental setup and data processing at NCB were the same as those at SCB, we focus on SCB in this part. The HR–ToF–AMS (Aerodyne) was deployed in an air-conditioned room with the temperature maintained at about 24 ℃ at SCB to measure particulate matter, typically referred to as non-refractory $PM_1$ (Canagaratna et al., 2007). A $PM_{2.5}$ cyclone inlet was set up on the roof of the building to remove coarse particles and introduce the air stream into the room through a copper tube with a flow rate of $10\ l\ min^{-1}$. The HR–ToF–AMS sampled isokinetically from the center of the

10   copper tube at a flow rate of $80\ ml\ min^{-1}$. In addition, a nafion dryer was positioned upstream of the HR–ToF–AMS to eliminate the potential influence of relative humidity on particle collection (Matthew et al., 2008). A more detailed technical description of the HR–ToF–AMS can be found in the literature (DeCarlo et al., 2006; Canagaratna et al., 2007). The HR–ToF–AMS was calibrated for inlet flow, particle sizing, and ionization efficiency (IE) at the beginning and end of the campaign by following the standard protocols (Drewnick et al., 2005; Jayne et al., 2000). The IE and size calibration were

15   performed with size-selected pure ammonium nitrate particles. The instrument was operated in two ion optical modes in a cycle of 4 min, including 2 min for the Vmode, to obtain the mass concentrations of the non-refractory species, in which there was 100 s of time for the particle time-of-flight (PToF) mode to determine the size distributions of the species, and 2

min for the W mode to obtain high-resolution mass spectral data. An aethalometer (AE-31, Magee) was used for simultaneous measurement of refractory black carbon (BC) with a time resolution of 5 min for a better closure of fine particles in the campaign.

### 2.3. HR–ToF–AMS data processing

5 The HR–ToF–AMS data analysis was performed using the standard AMS data analysis software packages (SQUIRREL version 1.57 and PIKA version 1.16, downloaded from the ToF–AMS–Resources webpage1) in Igor Pro 6.37 (Wave Metrics Inc.). The composition-dependent collection efficiency (CE) was applied to the data based on the method in Middlebrook et al. (2012), and the obtained CE values were mostly around 0.5. Organic elemental analysis was carried out using the latest procedures (Canagaratna et al., 2015), which improved the estimation method from that of Aiken et al. (2008).

10       PMF analysis has been widely applied in AMS data processing for organic aerosol source apportionment (Zhang et al., 2010; Ng et al., 2010; Huang et al., 2010, 2013). In this study, PMF analysis was performed on the high-resolution organic mass spectra (m/z 12-150) obtained from HR–ToF–AMS data. The organic data matrix and error matrix input into the PMF analysis (Ulbrich et al., 2009) were generated with the default fragmentation waves in PIKA version 1.16. Before running the PMF, ions with a signal-to-noise ratio less than 0.2 were removed and ions with a signal-to-noise ratio ranging between 15 0.2 and 2 were downweighted by a factor of 2. The ions of $H_2O^+$, $HO^+$, $O^+$, and $CO^+$ were removed from the data and error matrices since they were determined according to their relationship with $CO_2^+$ and thus including them in the PMF analysis could introduce additional weight to $CO_2^+$ (Ulbrich et al., 2009). The obtained PMF solutions were evaluated based on the procedure outlined in Zhang et al. (2011). The optimal solutions at both sites were examined for their residuals of PMF fits, MS signatures, their correlation with tracers, diurnal variations, and other characteristics. The diagnostic plots of the chosen 20 results at the two sites are shown in Figures S1 and S2 in the supporting information. The results show that PMF solutions with factor number greater than three at NCB and two at SCB provided no new distinct factors and instead displayed splitting behavior of the existing factors. The $Q/Q_{expected}$ and the factors obtained for different FPEAK (from −1.0 to 1.0) and SEED (from 0 to 250) values made a small difference in the OA components produced. An FPEAK value of 0 was finally used for both sites because of the lowest $Q/Q_{expected}$ and that the use of FPEAK values different from 0 did not improve the 25 correlations between PMF factors and external tracers. Based on all tests in the PMF analysis, the three factors at NCB were assigned as HOA, OOA1, and OOA2, and the two factors at SCB were identified as SV-OOA and LV-OOA, as further discussed in Section 3.5. More cases of the PMF analysis can be found in our previous publications (Huang et al., 2010; He et al., 2011; Huang et al., 2012).

### 2.4. Estimation of organic nitrates

30 Note that the nitrate measured by the HR–ToF–AMS is the nitrate functionality (-$ONO_2$), which may include inorganic and organic nitrates. Farmer et al. (2010) reported an approach to quantify organic nitrates with the HR–ToF–AMS data. The

nitrate portion of inorganic and organic nitrates primarily fragments to $NO^+$ and $NO_2^+$ ions. Previous laboratory studies have shown that $NO^+/NO_2^+$ ($NO_X^+$ ratio) is different for organic nitrate and $NH_4NO_3$, and several groups have reported that the $NO_X^+$ ratios observed in the AMS spectra for organic nitrates are typically 2–3 times higher than those for $NH_4NO_3$ (Fry et al., 2009; Bruns et al., 2010; Farmer et al., 2010; Boyed et al., 2015). Due to the very different average $NO_X^+$ ratios ($R_{NH4NO3}$ and $R_{ON}$), the $NO_2$ and $NO$ concentrations for ON ($NO_{2,ON}$ and $NO_{ON}$) can be estimated by Eqs. (1) and (2):

$$NO_{2,ON} = \frac{NO_{2,obs} \times (R_{obs} - R_{NH4NO3})}{R_{ON} - R_{NH4NO3}} \tag{1}$$

$$NO_{ON} = R_{ON} \times NO_{2,ON} \tag{2}$$

where $R_{obs}$ is the ambient $NO_X^+$ ratio. For each dataset, $R_{NH4NO3}$ is determined by IE calibration using pure $NH_4NO_3$ at the beginning and the end of the campaigns and the results show stable values: at NCB, the calibration $R_{NH4NO3}$ was 2.68 and 2.59 at the beginning and the end of the campaign, respectively, and that was 3.40 and 3.20, respectively, at SCB. We thus chose the average of the two IE calibrations as the $R_{NH4NO3}$ for each campaign, which was 2.61 ±0.13 and 3.28 ±0.17 at NCB and SCB, respectively, well within the range in the literature (Sato et al., 2010; Fry et al., 2013). The value for $R_{ON}$ is more difficult to determine because it varies between instruments and precursor VOCs. Fry et al. (2013) assumed that $R_{ON}/R_{NH4NO3}$ is instrument-independent, and further calculated its value to be 2.25 ± 0.35 based on Farmer et al.'s (2010) results. However, only a few $R_{ON}/R_{NH4NO3}$ values have been reported in the literature so far, such as 2.25± 0.35 for the organic nitrate standard (Farmer et al., 2010), 3.99± 0.25 for the organic nitrates produced from beta-pinene (Boyed et al., 2015), and 2.08± 0.14 for isoprene (Bruns et al., 2010). Sato et al. (2010) showed that the $R_{ON}/R_{NH4NO3}$ for the organic nitrate through photo-oxidation of aromatics is about 2.45. Thus, only a $R_{ON}/R_{NH4NO3}$ estimation range (from 2.08 to 3.99) can be defined from the literature due to the variation of precursor VOCs. In this study, the small difference between $R_{obs}$ (2.79±0.63) and $R_{NH4NO3}$ (2.61 ±0.13) at NCB indicated a limited fraction of organic nitrates in the total nitrate with consideration of the uncertainty. However, at SCB, $R_{obs}$ (5.39± 1.73) was significantly higher than $R_{NH4NO3}$ (3.28± 0.17), implying significant existence of organic nitrates. A low measured/predicted $NH_4^+$ ratio (~0.94) at SCB was another implication for the presence of organic nitrates. In contrast to the abundant existence of metal nitrates in coarse mode particles (Huang et al., 2006), e.g., $NaNO_3$, one may assume low contributions of metal nitrates in AMS detection in the submicron range. Thus, the lower and upper limits of $R_{ON}$ were calculated to be 6.82 and 13.08, respectively, to estimate the amount of organic nitrates at SCB in Section 3.2.

## 2.5. TPSCF analysis

PSCF analysis is a receptor model that explicitly incorporates meteorological information in the analysis scheme to produce a probability field for source emission potential. Cheng et al. (1993) improved the method by considering the influence of different height layers, called TPSCF analysis. In this study, air-mass back-trajectories from the previous 48 h were determined using the Hybrid Single-Particle Lagrangian Integrated Trajectory model (version 4.9; Draxler and Rolph, 2003) at five different endpoint heights (100 m, 200 m, 300 m, 400 m, and 500 m) and a time interval of 1 h for each day. After

that, the geographic region covered by the trajectories was divided into an array of 0.5×0.5 grid cells. The number of the trajectory segment endpoints over the $ij$th grid cell for height $k$ is counted as $n_{ij}^k$. The number of these endpoints corresponding to a pollutant concentration higher than a criterion value for height $k$ is counted as $m_{ij}^k$. The mean value for each species was set as the pollution criterion. In order to reduce the effect of small values of $\sum n_{ij}^k$, the TPSCF values were multiplied by an arbitrary weight function $W_{\sum n_{ij}^k}$ to better reflect the uncertainty in the values for these cells (Kedia et al., 2012; Wu et al., 2009; Zhu et al., 2011). In this case, we use the power of the number of trajectories ($T$) at one endpoint height to determine the categories of $\sum n_{ij}^k$, and $W_{\sum n_{ij}^k}$ was defined as below (Guo et al., 2015):

$$W_{\sum n_{ij}^k} = \begin{cases} 1.00, T^{0.7} < \sum n_{ij}^k \\ 0.70, T^{0.56} < \sum n_{ij}^k \leq T^{0.7} \\ 0.42, T^{0.42} < \sum n_{ij}^k \leq T^{0.56} \\ 0.05, \sum n_{ij}^k \leq T^{0.42} \end{cases} \tag{3}$$

## 3. Results and discussion

### 3.1. PM$_1$ composition and size distribution

Figures 2a and 2b show the time series of the PM$_1$ mass concentrations of different components during the two campaigns, respectively. The average PM$_1$ mass concentration (sum of the measured species) was $36.8 \pm 19.8$ μg m$^{-3}$ (mean ± standard deviation) at NCB (varying from 2.9 to 111.1 μg m$^{-3}$), which is much higher than that observed at SCB ($10.9 \pm 7.8$ μg m$^{-3}$, varying from 0.1 to 41.6 μg m$^{-3}$). It indicated that there was more severe aerosol pollution in the northern region than in the southern region in China. The chemical composition of PM$_1$ was dominated by OA (27.2%), nitrate (26.7%), and sulfate (22.0%) at NCB, while at SCB, OA (43.5%) and sulfate (30.5%) were the two most abundant PM$_1$ components of the total mass, as shown in Figures 2c and 2d. Nitrate only contributed 4.7% to the total PM$_1$ mass at SCB, suggesting that NO$_X$ emissions had a minor influence here in comparison with that at NCB. In addition, the higher ambient temperature (as shown in Table 1) at SCB would lead to more aerosol-to-gas partitioning of the semi-volatile nitrate. Note that the BC mass fractions in PM$_1$ in Figures 2c and 2d are likely to be overestimated because BC was measured for PM$_{2.5}$ by an aethalometer. This overestimation could be less than 20% according to the ambient BC size distributions measured at an urban site in South China (Lan et al., 2011).

Figures 2e and 2f represent the variation in relative contributions of different species as a function of the total $PM_1$ mass concentration at the two background sites. It is found that nitrate shows a continuously increasing fraction when $PM_1$ was accumulating at NCB, while both sulfate and OA played an important role when $PM_1$ was accumulating at SCB. This implies that $NO_X$ emissions played a critical role in regional air pollution in the more polluted northern China.

5    In terms of particle size distribution, all species at the two background sites generally show a similar accumulation mode, with a mass median diameter (MMD, vacuum aerodynamic diameter) at a large size of ~550 nm in vacuum aerodynamic diameter ($D_{va}$) (Figures 3a and 3b), which indicated very aged and internally mixed aerosol particles (Jimenez et al., 2003; Huang et al., 2010). In comparison to the size distribution patterns observed previously in urban atmospheres in China, the size distributions of OA at the two sites had a much smaller mass "hump" around 200−300 nm, which typically represents

10    freshly emitted particles (Huang et al., 2010; He et al., 2011; Huang et al., 2012). These size distribution patterns indicated little fresh local emissions at the two background sites. Figures 3c–3h show the variation in normalization size distribution as a function of component mass concentration. With the mass concentration increasing, the size distributions of OA, sulfate, and nitrate at the two sites become concentrated in larger sizes, further confirming that the high pollution levels there were determined by the regional transport of aged particles.

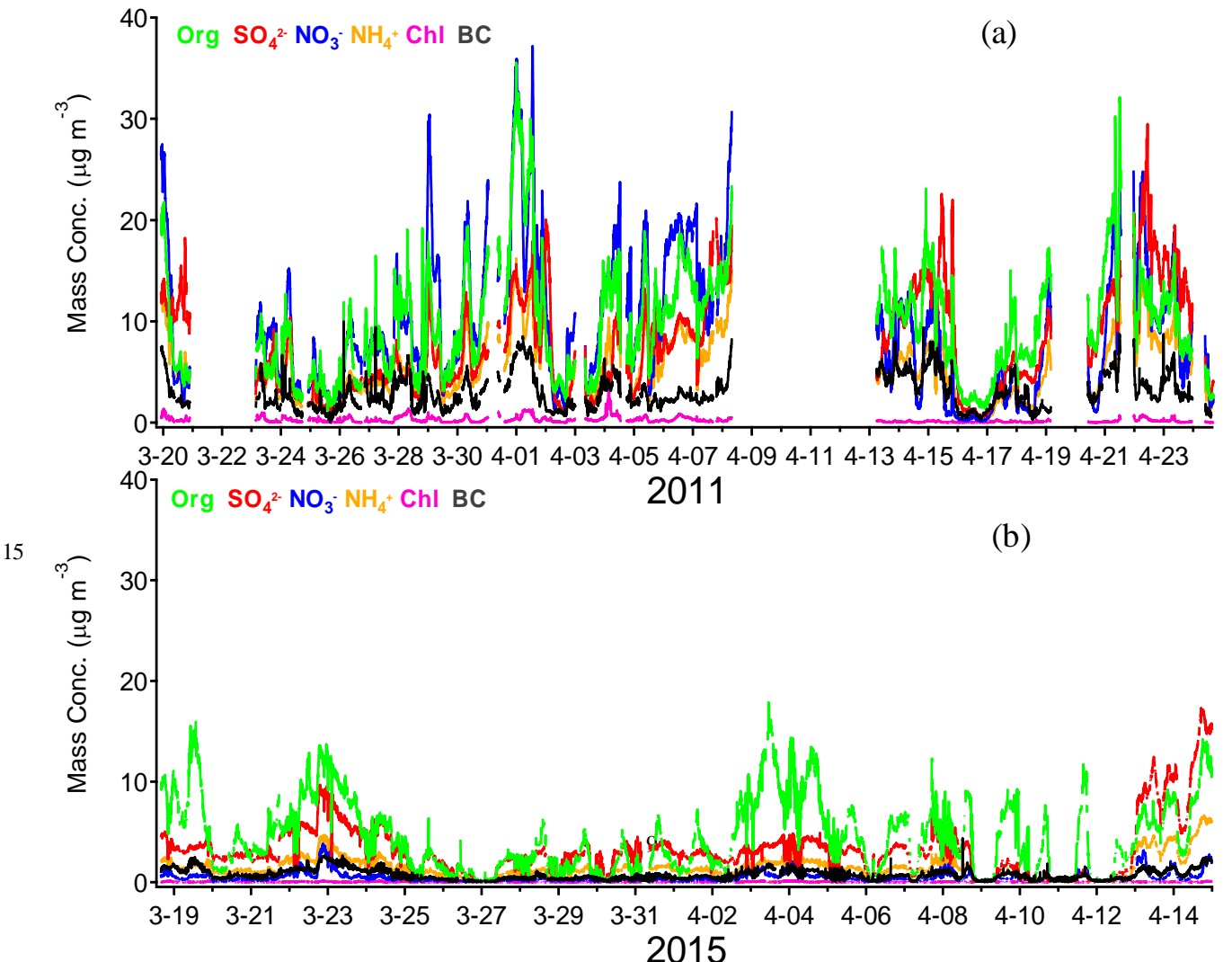

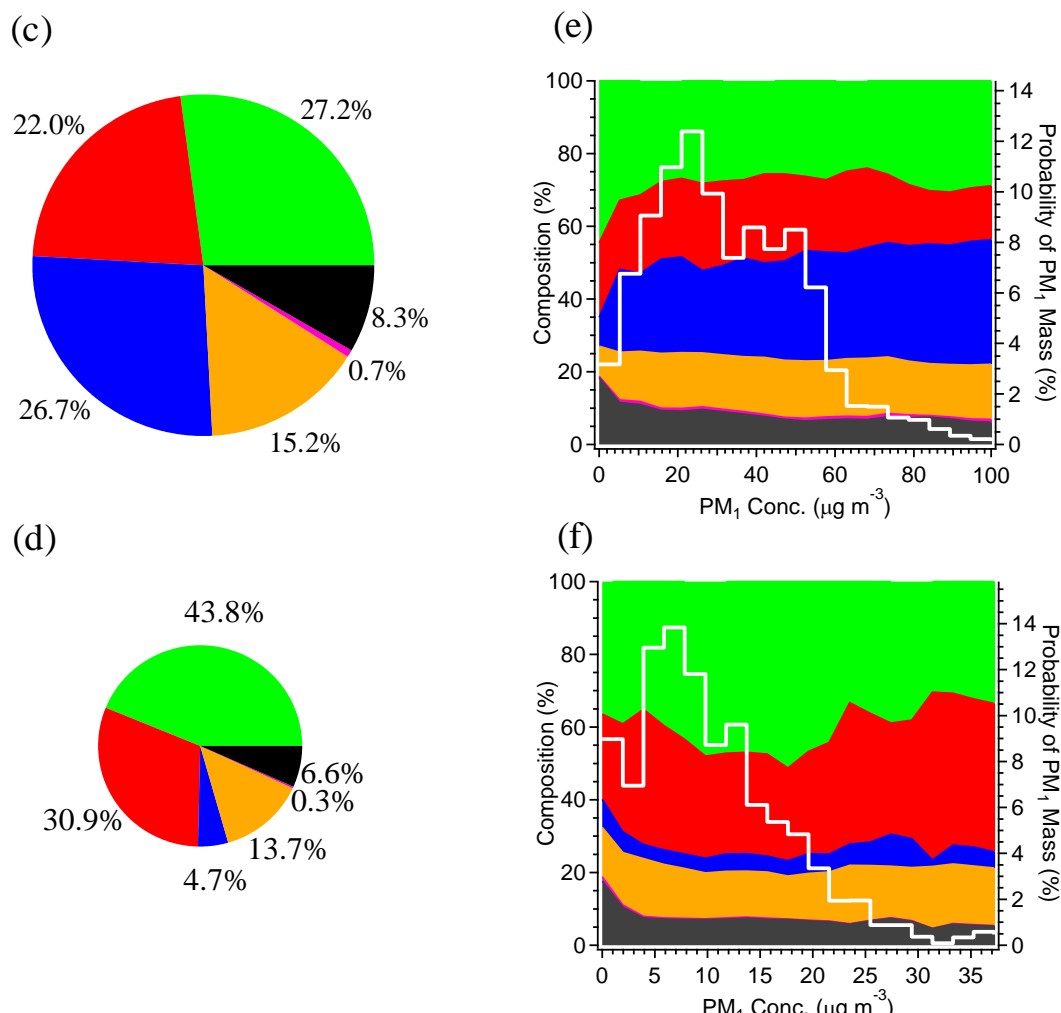

**Figure 2.** Time series of AMS species and BC at (a) NCB and (b) SCB; the average PM$_1$ chemical compositions at (c) NCB and (d) SCB, the areas of pie charts are sized by PM$_1$ mass loading; evolutions of PM$_1$ compositions (left axis) as a function of PM$_1$ mass loading, and the probability distributions of PM$_1$ mass loading (white line to right axis) at (e) NCB and (f) SCB.

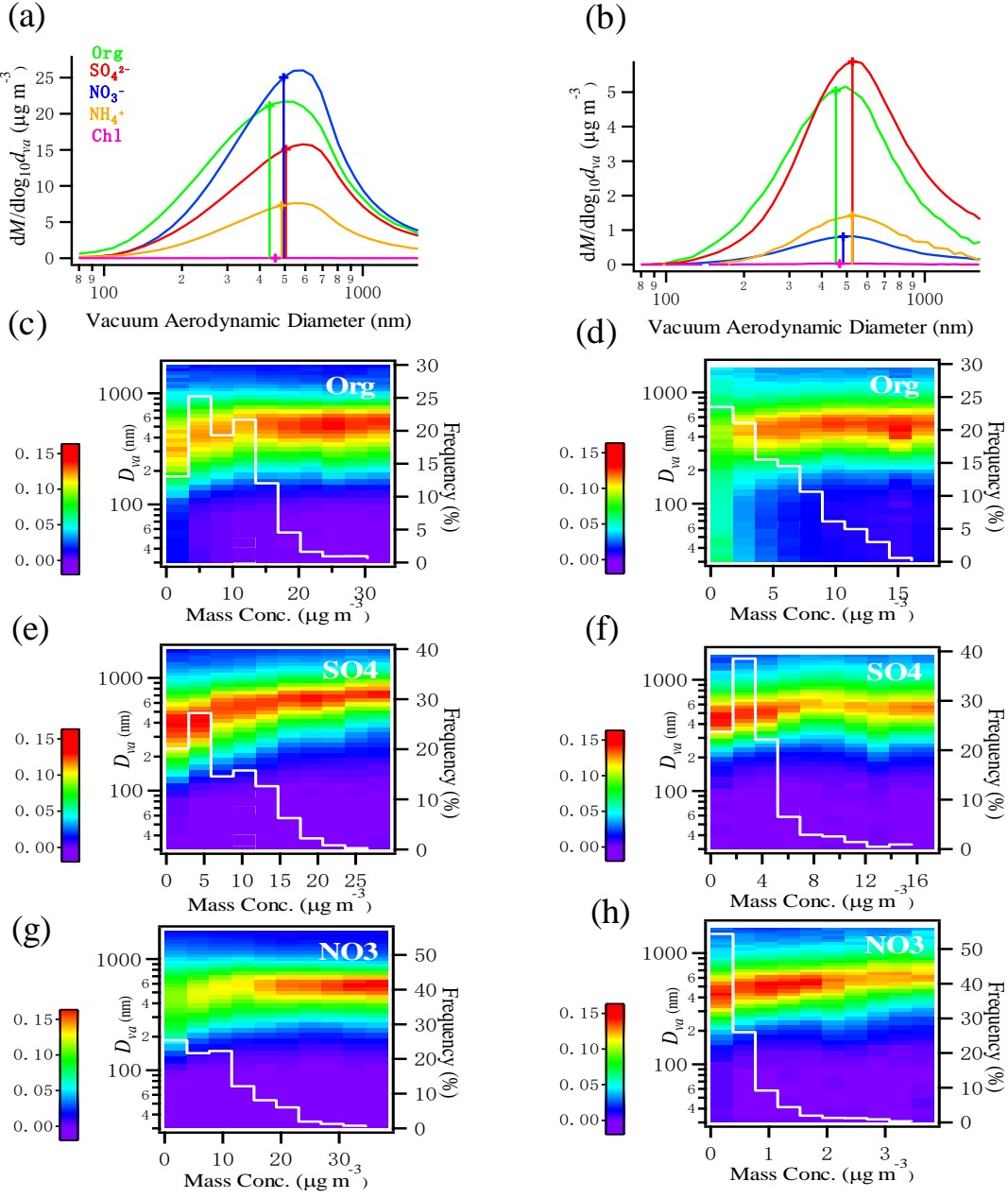

**Figure 3.** Average size distributions of AMS species, together with MMD (vacuum aerodynamic diameter) at (a) NCB and (b) SCB; variation in normalized size distribution as a function of component mass concentration at (c, e, g) NCB and (d, f, h) SCB.

## 3.2. Estimation of organic nitrates

Organic nitrates are a less-explored but potentially important component of secondary organic aerosol. Most atmospheric organic nitrates are produced either by photochemical (OH-initiated) or nocturnal ($NO_3$-initiated) oxidation reactions of anthropogenic and biogenic volatile organic compounds (Farmer et al., 2010). Recently, several direct and indirect approaches have been proposed to estimate organic nitrates from direct measurement data (Rollins et al., 2012; Farmer et al., 2010; Xu et al., 2015b). Xu et al. (2015b) estimated organic nitrates based on HR–ToF–AMS data and showed that organic nitrates contribute about 5–12% of the total OA for summer datasets and 9–25% of the total OA for winter datasets in the southeastern USA. However, similar studies on the contribution of organic nitrates to total OA in China have not yet been seen. As noted in Section 2.4, estimation of organic nitrates at NCB was not plausible using the $NO^+/NO_2^+$ method because of the low $NO_X^+$ ratios observed for the ambient aerosol, which indicated that organic nitrates were negligible in comparison with the large amount of inorganic nitrate (Farmer et al, 2010) At SCB, the ambient $NO^+/NO_2^+$ ratio ($R_{obs}$) fluctuated well within the range from $R_{NH4NO3}$ (3.28) calibrated to the maximum $R_{ON}$ (13.08), allowing for the feasibility of organic nitrates estimation at SCB. Although the concentrations of organic nitrates at NCB might not be smaller than at SCB, it is harder to quantify organic nitrates at NCB because ammonium nitrate at NCB was much higher than at SCB so that the uncertainties with this method are much higher. The results obtained using the $NO_X^+$ ratio method described in Section 2.4 are presented in Table 2. The mass fraction of the nitrate functionality from organic nitrates (i.e., $NO_{3,ON}/NO_{3,obs}$) was found to be 15–22% at SCB. Furthermore, by assuming that the average molecular weights of particle-phase organic nitrates are $200-300\ g \cdot mol^{-1}$ (Rollins et al., 2012), the calculation indicated that organic nitrates contributed 5−11% to the total OA at SCB. These results indicated that organic nitrates existed significantly in aerosol particles at SCB.

**Table 2.** Results of organic nitrates estimated using the $NO_X^+$ ratio method.

| Site | $R_{NH4NO3}$ | $R_{obs}$ | $NO_{3,ON}$ Conc. ($\mu g\ m^{-3}$) | | $NO_{3,ON}/NO_{3,obs}$ | | ON/OA | |
|------|--------------|-----------|-------|-------|-------|-------|-------|-------|
| | | | lower | upper | lower | upper | lower | upper |
| NCB | 2.61 ± 0.13 | 2.79 ± 0.63 | - | - | - | - | - | - |
| SCB | 3.28 ± 0.17 | 5.39 ± 1.73 | 0.08 | 0.12 | 15% | 22% | 5% | 11% |

## 3.3. Diurnal patterns of $PM_1$ species

The mean diurnal variations in $PM_1$ components at the two background sites are shown in Figure 4. BC at NCB shows lower concentrations in the daytime, while it shows higher concentrations in the daytime at SCB. This should be attributed to the different altitudes of the two sites: when the planetary boundary layer (PBL) is uplifted in the daytime, near-ground pollutants are transported vertically to the upper atmosphere, and thus, the BC concentrations at high attitudes increase, while those near the ground decrease. The small BC peak in the early morning at NCB could be attributed to the influence of

highway traffic emissions in a local scale, which increased quickly in the early morning. The diurnal patterns of OA, nitrate, sulfate, ammonium, and chloride generally show similar trends to those of BC at NCB, mainly driven by the diurnal variation in PBL. At SCB, OA shows much higher daytime concentrations in comparison with the diurnal pattern of BC due to photochemical formation, as supported by the simultaneous trend of the secondary species, that is, organic nitrates (in Figure 4), while the secondary formation of nitrate seemed not to compensate its mass loss due to the particle-to-gas vaporization of $NH_4NO_3$ during the daytime. Note that, at either NCB or SCB, the diurnal variation in sulfate was not as great as the others, suggesting that sulfate was well mixed through the boundary layer.

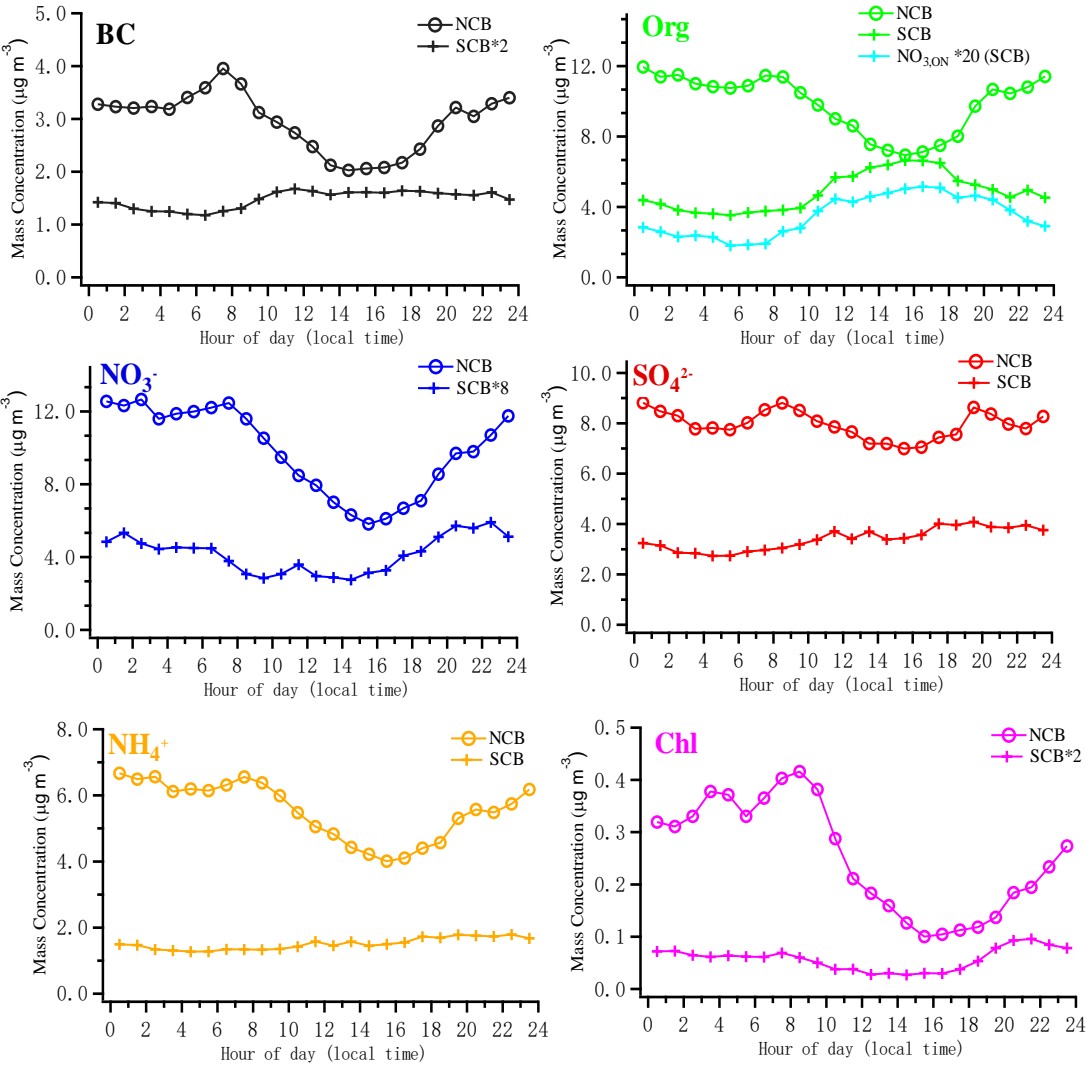

**Figure 4.** Diurnal variations in $PM_1$ components at NCB and SCB.

## 3.4. Organic aerosol aging process

Figure 5 shows the Van Krevelen diagrams (H/C versus O/C) of OA at NCB and SCB. The average O/C ratio at SCB was 0.98, which was much higher than that at NCB (0.67), indicating that OA in southern China was generally more oxidized and aged. Some campaign-average O/C and H/C ratios observed at urban sites in China by our group (Huang et al., 2010; Huang et al., 2012; He et al., 2011) and at remote/background sites in the literature (Chen et al., 2009; Chen et al., 2015; Robinson et al., 2011) are also plotted in Figure 5, and all the values in the literature have been corrected by the "Improved-Ambient" method described in Canagaratna et al. (2015). Noted that both O/C and H/C calculated with the Improved-Ambient method (Canagaratna et al.2015) should be higher than those using the previous method (Aiken et al., 2008) due to the underestimation of intensities of the $H_2O^+$ and/or $CO^+$ fragments by the previous method. A comparison of the elemental ratios in our previous campaigns in China between the previousand "Improved-Ambient (I-A)" methods can be found in Table S1 in the supporting information.The O/C and H/C ratios at SCB and NCB were found to be closer to those at other remote/background sites but further from those at the urban sites, consistent with SCB and NCB being background sites. Heald et al. (2010) proposed using the Van Krevelen diagram to illustrate how reactions involving addition of functional groups fall along straight lines for ambient aerosol. Although many other factors, such as the mixing of different air masses and components/sources, may also lead to a variety of slopes in the Van Krevelen diagram in the case of ambient field measurements, the Van Krevelen diagram may still be useful for constraining reactions that are responsible for the aging of OA (Hayes et al., 2013). The O/C versus H/C points at the two sites were fitted using the reduced-major-axis regression method (Smith, 2009), which is considered to be the best fit for a bivariate relationship when the variable represented on the $x$-axis is measured with error. The fitted lines at NCB and SCB agreed with each other very well and have a common slope of about −0.7, implying that the continuous oxidation processes of OA were controlled by mixed mechanisms, such as the formation of carboxylic acids with or without fragmentation, alcohol, and carbonyl addition (Heald et al., 2010), and would follow this empirical line in the Van Krevelen diagram in both northern and southern China. Note that the organic aerosol observed at SCB is very highly oxygenated compared to the ambient data ever reported in the literature, consistent with the previous finding that the atmospheric oxidizing capacity in southern China is unexpectedly high (Hofzumahaus et al., 2009).

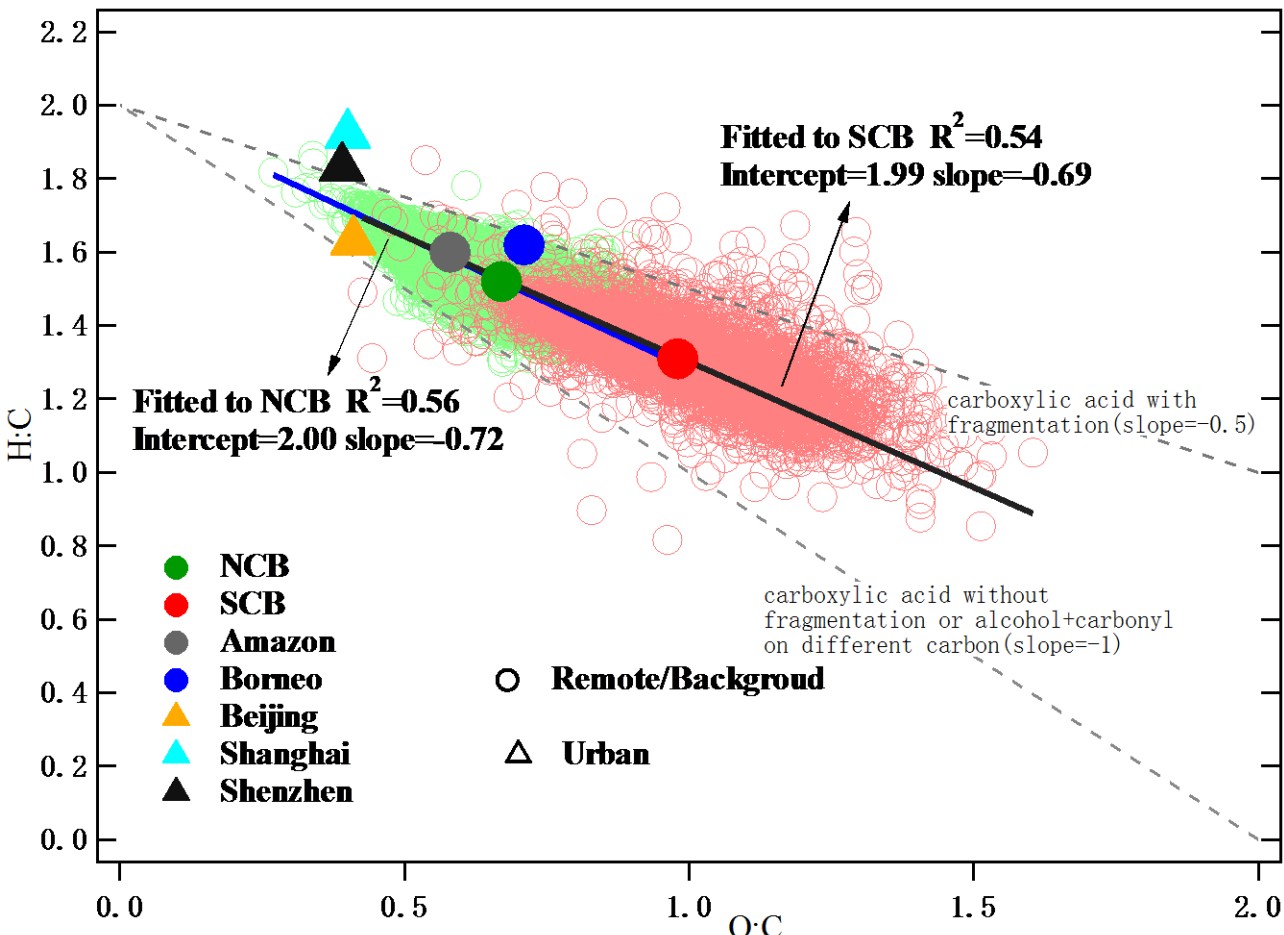

**Figure 5.** Van Krevelen diagrams (H/C versus O/C) of OA at NCB (light green) and SCB (light red). The campaign-average O/C and H/C ratios of remote/background and urban sites in the literature are also included.

### 3.5. Organic aerosol source apportionment

5    Three organic components, including a hydrocarbon-like OA (HOA) and two oxygenated OA (OOA1 and OOA2), at NCB were identified using the PMF method described in Section 2.3. While at SCB, only two oxygenated OA components (SV-OOA and LV-OOA) were identified. In this case, the contribution of primary organic aerosol (POA) might be too small for PMF to identify. At a remote site in a boreal forest in Finland and a rural site in the southeastern USA, no POA factor was identified by PMF either (Raatikainen et al., 2010; Xu et al., 2015a).

10       Figures 6a and 6b show the MS profiles of the OA factors and Figures 6c and 6d present their time series during the two campaigns. At NCB, HOA, OOA1, and OOA2 on average accounted for 30.6, 48.6, and 20.8% of the total OA mass, respectively. The HOA factor with a low O/C ratio of 0.28 was primarily dominated by the ion series of $C_xH_y^+$ and showed a good correlation with BC ($R^2$=0.51), indicating that it was directly emitted from fossil fuel combustion (Zhang et al., 2005;

Jimenez et al., 2009). Note that $f44$ is a little bit higher than $f28$ in the MS profile of HOA, suggesting that HOA is likely mixed with some OOA that cannot be separate by PMF and thus the actual contribution of HOA could be lower. The sum of OOA1 and OOA2 showed a high correlation with the sum of sulfate and nitrate ($R^2$=0.76, in Figure 6c), confirming their secondary nature. Although the two OOA components at NCB had the similar O/C ratios of 0.89 (OOA1) and 0.86 (OOA2),

their time series were significantly different, indicating their different source areas as discussed in Section 3.6. A similar case of OOA splitting was also found in the 2008 Beijing Olympic Games AMS dataset (Huang et al., 2010). At SCB, the OA factor with a higher $CO_2^+$ fraction and O/C is more oxidized and aged, and is thus referred to as LV-OOA, while the OA factor with a lower $CO_2^+$ fraction and O/C is less oxidized and fresher, and is thus referred to as SV-OOA (Jimenez et al., 2009; Ng et al., 2010). SV-OOA and LV-OOA on average contributed 39.3 and 60.7% of the total OA mass, respectively.

Although many previous AMS measurements at urban and rural sites (Lanz et al., 2007; Ulbrich et al., 2009; He et al., 2011; Huang et al., 2012) found that SV-OOA and LV-OOA correlated well with nitrate and sulfate, respectively, most likely due to their common volatile nature (Jimenez et al., 2009; Ng et al., 2010), these correlations were not significant enough at SCB ($R^2$=0.40 and 0.24, respectively). Since the SV-OOA at SCB is a more aged one with a higher O/C ratio in comparison with the SV-OOA factors in previous measurements, its semi-volatility might not be similar to that of nitrate. On the other hand,

the different source regions of LV-OOA and sulfate, as discussed in the next section, may explain their inconsistent temporal variations. Instead, it is very interesting to find that BC was correlated well with both SV-OOA and LV-OOA (both $R^2$=0.67), and even better with the total OA ($R^2$=0.72) at SCB, suggesting that the OOA precursors and BC should have the same sources. Considering the special location of the SCB site, the most likely sources of BC were biomass burning and ship emissions, as discussed in Section 3.6.

As shown in Figures 6g and 6h, the average diurnal patterns of the two OOA components at NCB show lower concentrations in the daytime while the two OOA components at SCB show higher concentrations in the daytime, mainly due to the influence of PBL, as discussed in Section 3.3. The small early morning peak of HOA at NCB was similar to BC, due to the influence of highway traffic emissions in a local scale. At SCB, LV-OOA shows a milder diurnal trend than SV-OOA, consistent with it being a more aged OA component and better mixed in PBL. In addition, the dramatic increase in

SV-OOA in the daytime at SCB is similar to that in organic nitrates (in Figure 4), implying that organic nitrates are mostly associated with SV-OOA and both of them are largely formed in the daytime.

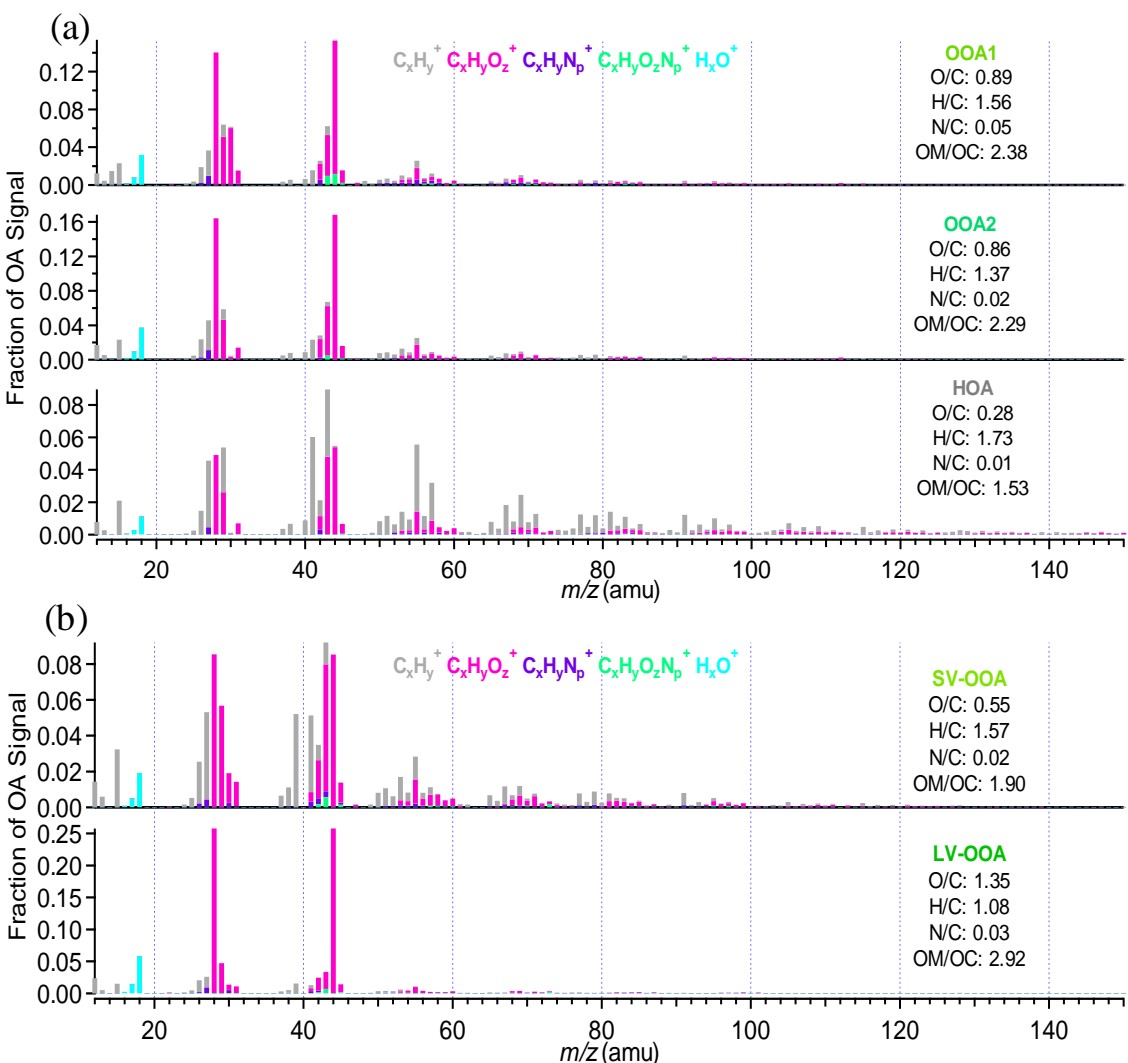

(c)

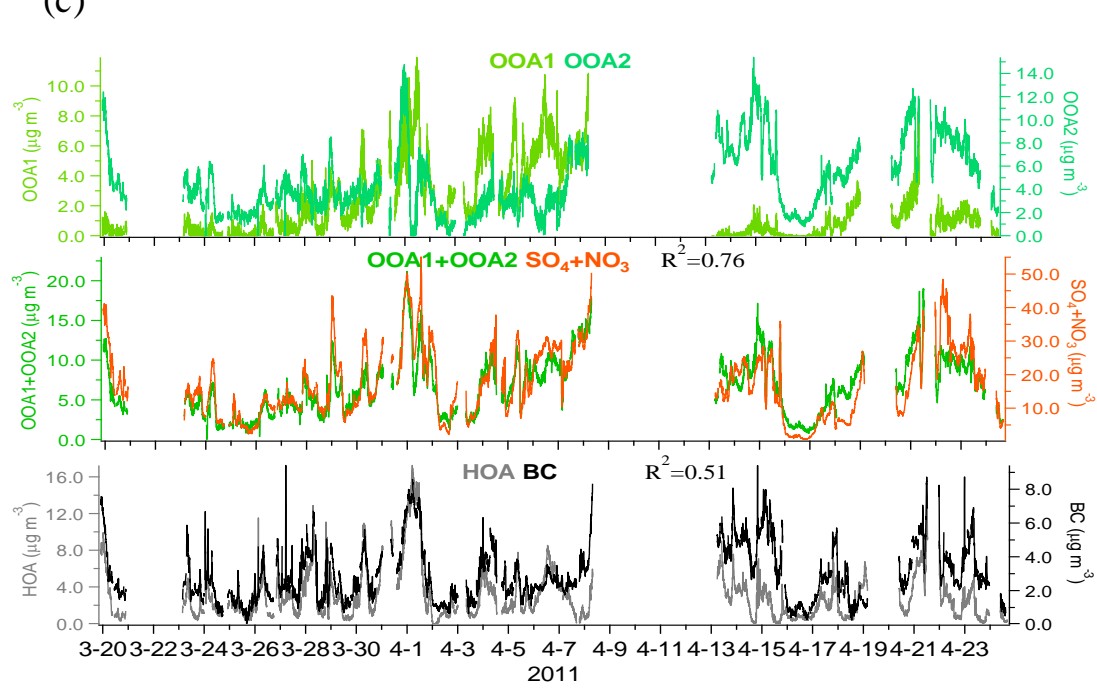

(d)

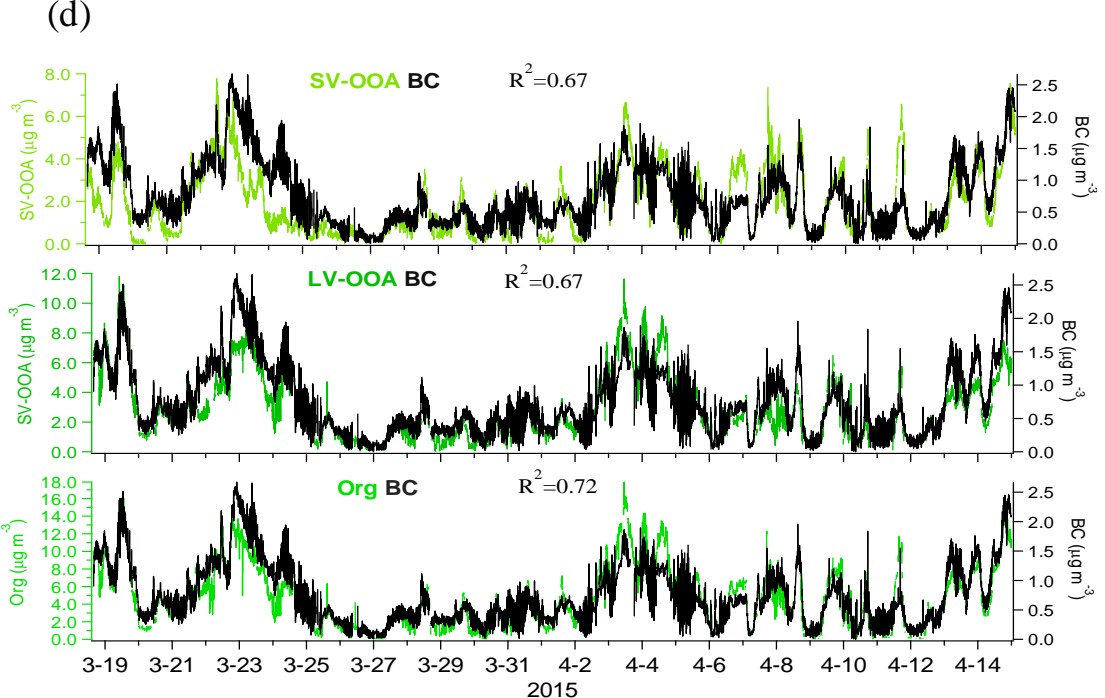

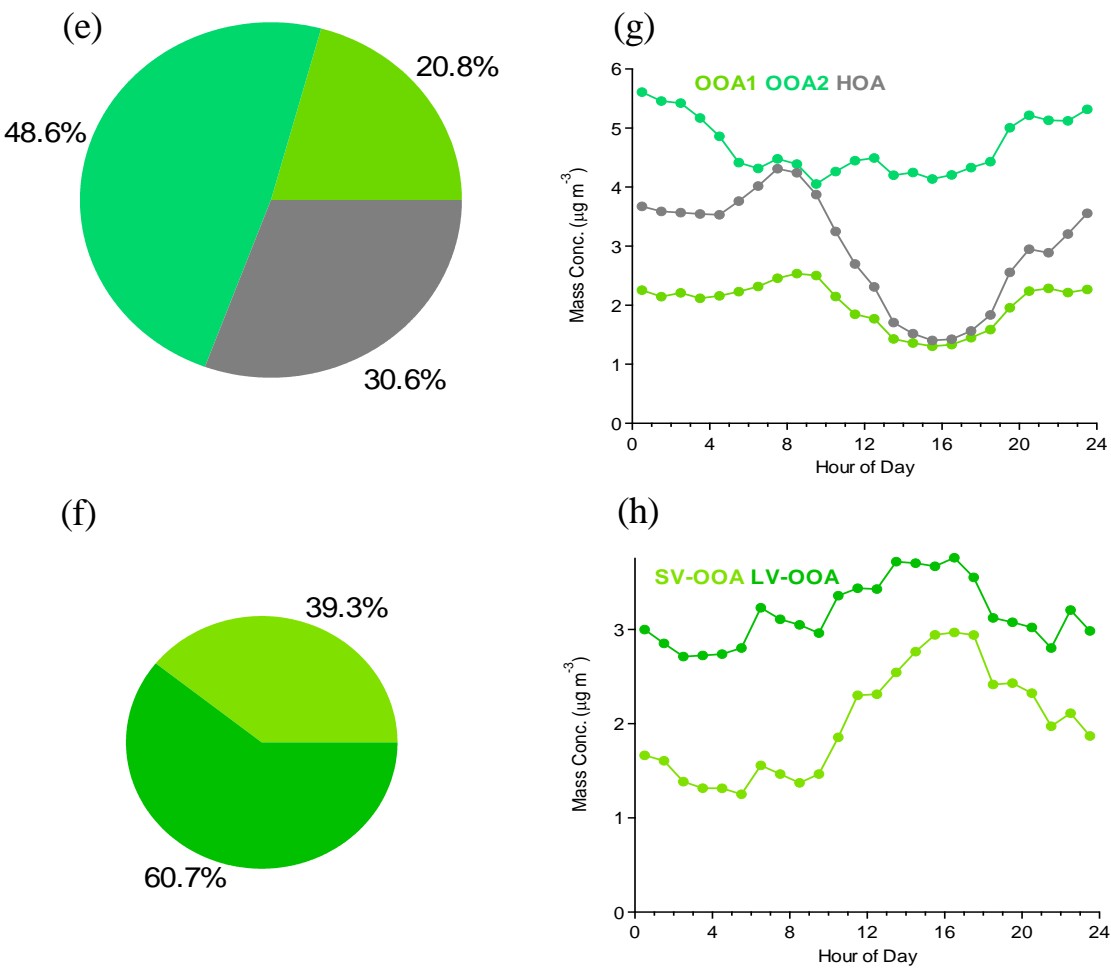

**Figure 6**. Mass spectrum profiles of the OA components at (a) NCB and (b) SCB identified by PMF. The time series of the OA components at (c) NCB and (d) SCB and other relevant species; the averages of the OA compositions at (e) NCB and (f) SCB; the diurnal patterns of the OA components at (g) NCB and (h) SCB.

### 3.6. Potential emission source areas analysis

The potential source areas identified by the TPSCF model for different aerosol components at both sites are shown in Figure 7. At NCB, the high-potential source region of BC was mainly located in the southern continental areas, including the highly urbanized and industrialized Yangtze River Delta (YRD) region and the region of the middle reaches of the Yangtze River. The HOA high potential source area was similar but smaller compared to that of BC, which is a reasonable result of the

poorer stability and thus shorter lifetime of HOA. For secondary species, including OOA, nitrate, and sulfate, besides the YRD region, the sea area to the east of YRD, extending to South Korea, also played an important role. In addition, we can clearly see that the high potential source areas of OOA1 and OOA2 were substantially different: OOA2 is mostly mainland-oriented, similar to HOA and BC, while OOA1 is uniquely from the sea area between China and South Korea. Ship emissions are reasonably assumed to be the responsible source on the sea, since the YRD area has two of the 10 biggest ports in the world, that is, the Shanghai port and the Ningbo-Zhoushan port. The engines of ocean ships typically consume heavy fuel oils, emitting huge amounts of $NO_X$, $SO_2$, VOCs, and particles, and are thus regarded as an important source of air pollution in a regional scale (Eyringa et al., 2010).

At SCB, BC, OA, sulfate, and nitrate all have three similar potentially important source areas. The first one is the boundary area among Laos, Cambodia, and Thailand, where biomass burning emissions are intensive in the spring season (Lin et al., 2013); Chuang et al. (2014) recently attributed high levels of carbonaceous aerosol in the background of Taiwan in spring to biomass burning in the Indo-China peninsula. The second one is the sea area to the southeast of Vietnam, which is a key international water course for cargo ships going between East Asia and the West, which also contains the biggest harbor in southern Vietnam. The third one is the northern mainland area, mainly in the adjacent and economically developed Guangdong Province. Based on radiocarbon ($^{14}$C) measurements at another regional background site very close to our SCB site on the Hainan Island, Zhang et al. (2014) also found that a higher contribution of fossil sources could be attributed to emissions from Southeast China while an increase of non-fossil sources was associated with open biomass burning activities in Southeast Asian countries. In comparison with other species, it is found that sulfate had more potential source areas in inland China, where coal is intensively used and $SO_2$ emissions are high, rather than in the Southeast Asian countries. As at NCB, LV-OOA at SCB had more intensive source areas on the sea than SV-OOA, implying that ship emissions can produce a large amount of very aged OA after regional transport and serve as a big contributor to regional background aerosol levels.

In summary, it is seen that the potential pollution source areas of the two background sites in China are on a regional scale rather than on a local scale, consistent with their nature as background sites. This highlights the need of long-term AMS measurements for all seasons at multiple sites in East Asia in order to have a comprehensive understanding of aerosol sources and transport in this region. Crippa et al. (2014) have recently well shown the advantages of multiple-site AMS measurements in acquiring insights of organic aerosol sources in Europe. In addition, note that the source areas are located not only in the Chinese territory but also in other neighboring countries, and biomass burning and ship emissions are found to be the likely sources causing cross-boundary air pollutant transport from neighboring countries to China.

(a)

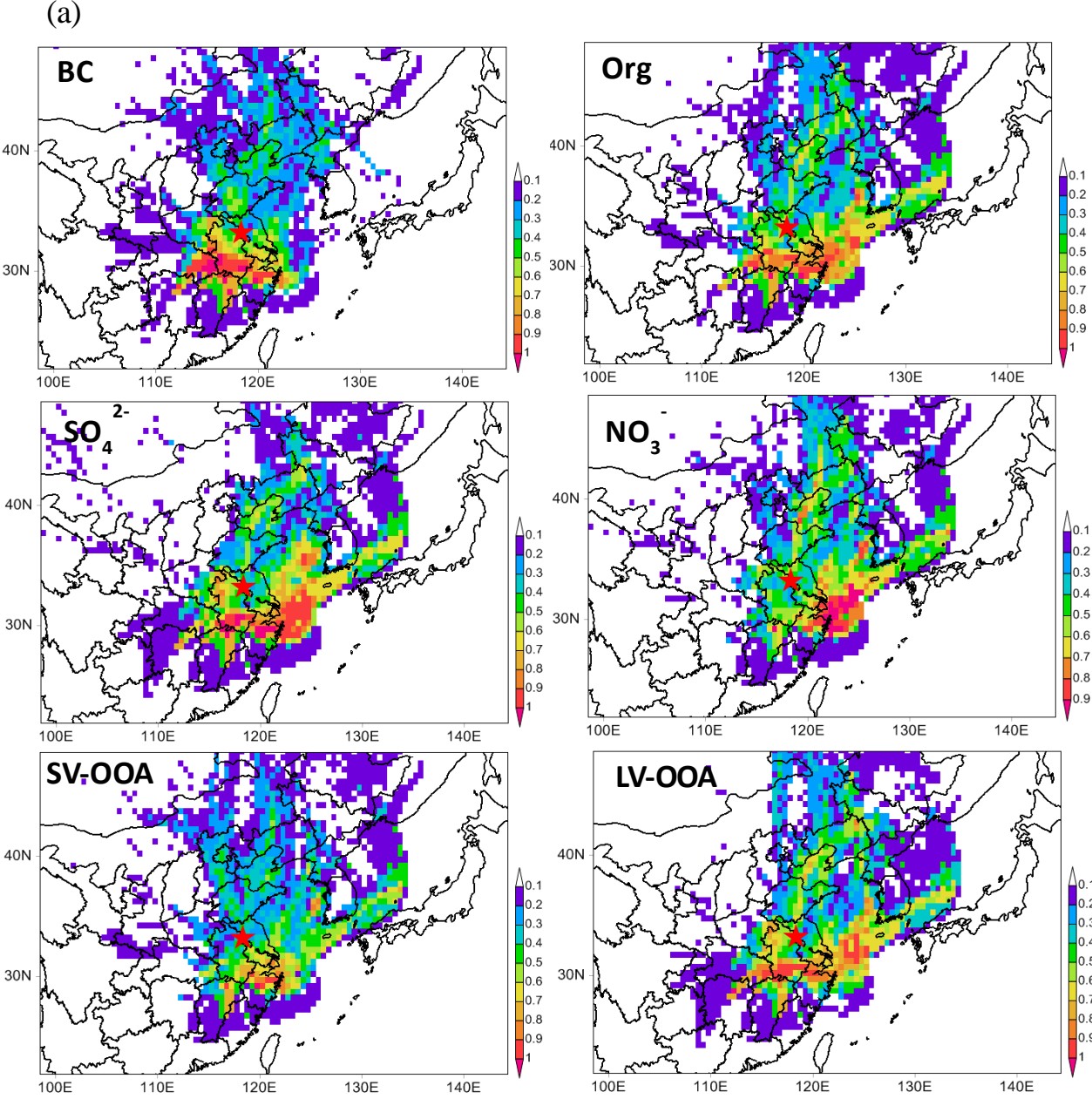

(b)

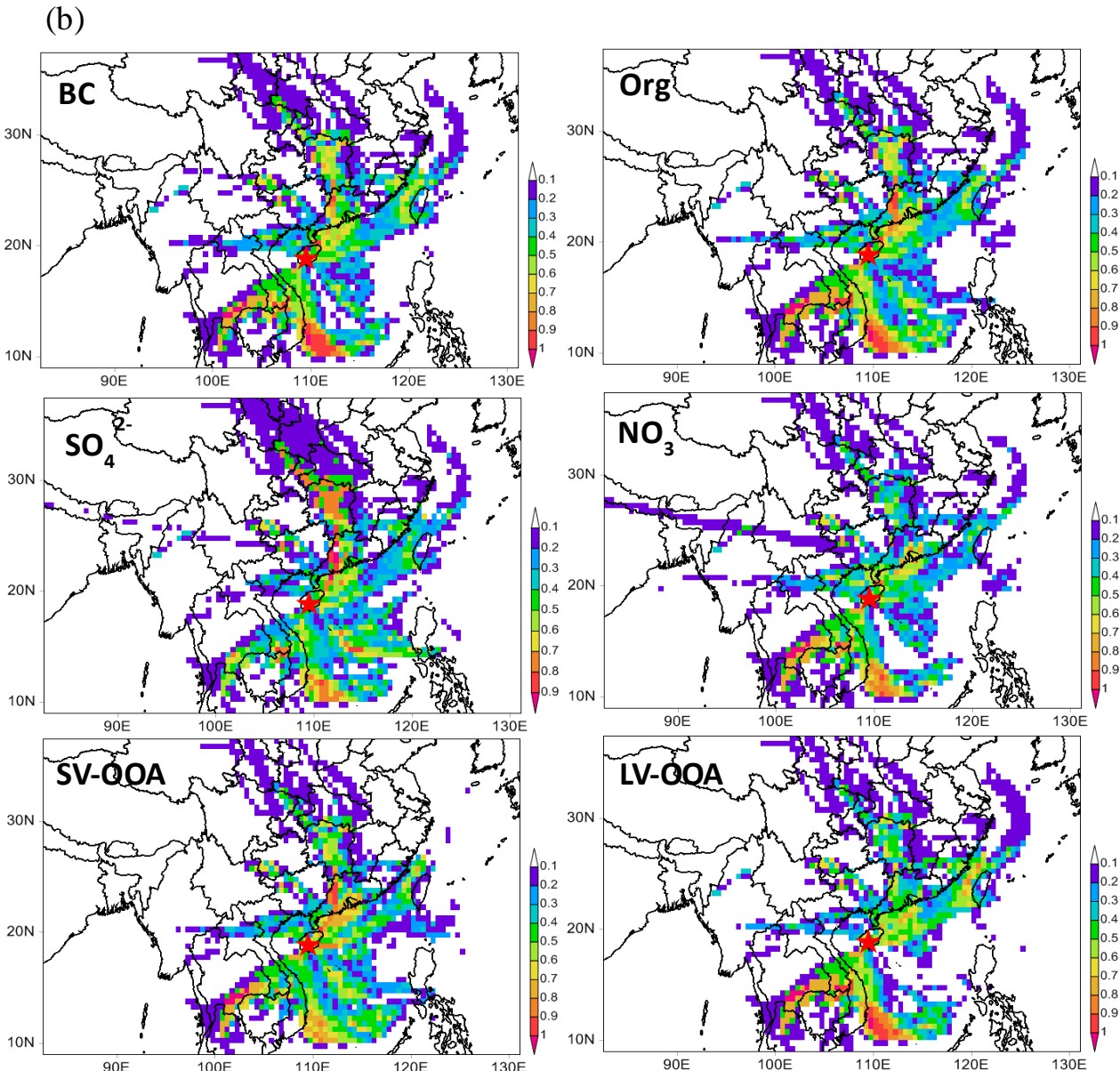

**Figure 7**. Mapping of the TPSCF results of major aerosol species at (a) NCB and (b) SCB.

## 4. Conclusion

Based on the HR–ToF–AMS measurements at two national background sites in southern and northern China during spring, the average PM$_1$ at NCB (36.8 ± 19.8 μg m$^{-3}$) was found to be much higher than that at SCB (10.9 ± 7.8 μg m$^{-3}$), suggesting more severe aerosol pollution in the northern part of China. At SCB, OA (43.5%) and sulfate (30.5%) were the most abundant PM$_1$ components, while nitrate contributed only a small fraction (4.7%). At NCB, however, nitrate accounted for a

comparable fraction (26.7%) of PM$_1$ mass to those of OA (27.2%) and sulfate (22.0%), revealing that northern China was much more influenced by NO$_X$ emissions. The average size distribution patterns of the species (except BC) at both sites were all dominated by an accumulation mode peaking at ~550 nm, indicating very aged particles. By using the NO$_X^+$ ratio method, organic nitrate was estimated to account for 15–22% of the total nitrate at SCB but only a negligible fraction at

NCB. OA at SCB was found to be more oxidized (O/C = 0.98) than that at NCB (O/C = 0.67) by organic elemental analysis, which was a reasonable result when considering the very high atmospheric oxidizing capacity in southern China.

PMF analysis of the high-resolution organic mass spectral data identified primarily-emitted HOA and secondarily-formed OOA1 and OOA2 with different source areas at NCB, while only secondary SV-OOA and LV-OOA at SCB. OOA2 (48.6%) and LV-OOA (60.7%) dominated the OA mass at NCB and SCB, respectively. The diurnal variation analysis suggested that

SV-OOA, likely containing a major part of organic nitrate, was largely formed in the daytime at SCB. TPSCF results of the major species of PM$_1$ at both NCB and SCB showed sources on a large regional scale. Besides mainland China, some territories of neighboring countries were also high-potential source areas, with biomass burning and cargo ship emissions as the most likely responsible sources for this cross-boundary air pollution.

**Acknowledgments**

This work was supported by the National Natural Science Foundation of China (21277003, 91544215) and the Ministry of Environmental Protection of China (Special Fund for Public Welfare 201309016).

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
