# Peer review of "Atmospheric aerosol compositions and sources at two national background sites in northern and southern China"

_Atmospheric Chemistry and Physics, 2016_

## Referee Comment (RC1) · Anonymous Referee #1 · 28 May 2016

The work of Zhu et al. reports HR-AMS measurements from two China sites. It reports the mass loading, chemical compositions, and PMF source apportioment results from ofãĂĂthe organics aerosols in both site. The paper is generally written clearly, but this reviewer finds a few major issues thus the paper may need major revision before its publication in ACP.

major comments (1)The introduction part is clearly not comprehensive and valuable. First, as the application of AMS in China increased significantly in recent years, the authors should do a bit more thourough summary of the current status; More recent studies should be mentioned, for example, AMS studies conducted in Beijing by Sun yele's group, in lanzhou ïijĹAtmos. Chem. Phys., 14, 12593-12611, 2014.ïijĽ, and a

more recent SP-AMS study in Nanjing (Environmental Science & Technology Letters, 3, 121-126,2016) and more. Secondly, if this paper intends to discuss the aerosol characteristics from background sites, the authors should summarize previous studies and major findings regarding the aerosol chemistry in background sites, and this should not be limited in china, but all over the world. What is the difference between the background sites and urban/polluted sites, and then what new findings do we expect in this manuscript?

(2)As there already are so many AMS papers published in the past 15 years, it is difficult to see what is the significance and novelty of this paper. This should be made more clear, the novelty should not because you did AMS measurements at sites that are different from others, but instead you should state what scientific questions and what valubale findings you gained from your measurements that can advance our current understanding on aerosol chemistry?

(3)As the two measurements were conducted at different years, they have very different meterological parameters, the discussion should consider and discuss more the meterological effects, while current version is clearly lacking of such discussion. No meterological data are shown. If so, the findings might be only for these two cases, having very limited scientific values for future and other studies.

(4)Regarding the quantification of organic nitrates, do the authors consider influences of metal nitrates? The AMS can measure nitrate that associated with sodium etc., although it is difficult to measure metals. Previous studies also pointed out that metal nitrates can have higher NO/NO2 ratios, so youe estimation method is incorrect without considering this point.

(5)The Vk diagram is quite limited in describing the formation processes of ambient OA in the reviewer's viewpoint. As there are so many possibilities that can influences the O/C and H/C ratios of ambient OA. The variation of O/C and H/C may not reflect the evolution processes at all. It maybe useful for chamber studies but should be discussed

with cautions for ambient data.

(6)Why a 3-factor solution was chosen for NCB site? It seems like 2-factor solution is also fine. In the supplement, it seems like the 2-factor solution is similar as the SCB site, Then why for NCB you chose 3-factor solution? Also, the Mass spectra of OOA1 and OOA2 in your 3-factor solution are quite similar, their diurnal patterns are similar too. Even if the authors insist to keep a POA factor for NCB site, this reviewer thinks OOA1 and OOA2 can be combined as one OOA factor.

A few technical comments (there maybe more, please check the MS carefully): page 2, line 15: "Valuable insights into the composition, sources, and evolution processes of was mostly found to be a submicron in China were obtained by the powerful on-line tools". What meaning? Please re-write this sentence.

Page 5, line 7: a flow rate of 80 l min$-1$? should be 80 ml min-1.

---

## Referee Comment (RC2) · Anonymous Referee #2 · 6 Jun 2016

The manuscript by Zhu et al. presented a general characterization of submicron aerosol composition, size distributions, and sources at two background sites (Lake Hongze site and Mount Wuzhi site) in China using high-resolution time-of–flight aerosol mass spectrometer. The results highlighted very different aerosol composition, diurnal variations, sources of organic aerosols, and elemental ratios at the two rural sites. Such two datasets are of interest and important for understanding aerosol chemistry at rural environments in China. I recommend a major revision before publication.

Comments:

1. It is difficult to compare the two studies directly since they were conducted in different years and different elevations. The authors should state clearly in the abstract that the two measurements were conducted in different years (2011 vs. 2015) and different elevations (21 m vs. 958 m) to avoid confusing the readers.
2. Calling "Lake Hongze" as a site in northern China is not accurate, in fact, "Central eastern China" might be better.
3. Abstract, the authors claimed "the most aged OA in real ambient air ever reported in the literature", which is not correct. Please refer to Chen et al. (Geophys. Res. Lett., 42, 4182–4189, 10.1002/2015GL063693, 2015).
4. Page 2, line 14, rewrite this sentence.
5. Page 2, line 29, I didn't quite understand why the regional background air pollution is a critical factor in determining urban air quality.
6. Page 3, line 1, "other instruments" actually refers to "AE31", I didn't see other collocated instruments.
7. All the names of submicron aerosol species should be synchronized. For example, "Cl⁻" vs Chl, "Organic in Figure 2" vs. "Organic aerosol", etc.
8. Page 5, line 7, flow is not correct.
9. Page 8, line 10, could you give a number for the overestimation?
10. Page 8, line 11,"Figure 2c"should be Figure 2e?
11. Figure 3a, the vertical lines did not match the maximum sizes.
12. Why the authors use different names for OOA components at the two sites, for example, OOA1 and OOA2 at Lake Hongze and SV-OOA and LV-OOA at Wuzhishan?
13. Figures 3a, add figure legend for aerosol species.
14. Page 3, line 8 and line 10, same latitude and longitude for the two sites?

---

## Author Comment (AC1) · 25 Jun 2016

1. The introduction part is clearly not comprehensive and valuable. First, as the application of AMS in China increased significantly in recent years, the authors should do a bit more through summary of the current status; More recent studies should be mentioned, for example, AMS studies conducted in Beijing by Sun yele's group, in Lanzhou Atmos. Chem. Phys., 14, 12593-12611, 2014., and a more recent SP-AMS study in Nanjing (Environmental Science & Technology Letters, 3, 121-126,2016) and more. Secondly, if this paper intends to discuss the aerosol characteristics from background sites, the authors should summarize previous studies and major findings regarding the aerosol chemistry in background sites, and this should not be limited in

china, but all over the world. What is the difference between the background sites and urban/polluted sites, and then what new findings do we expect in this manuscript? RE-PLY: We add more relevant descriptions in the introduction part. For more complete summarization of AMS studies in China, as below: Since 2006, valuable insights on the composition, sources, and evolution processes of submicron particles in China were obtained through a dozen of field campaigns using various types of Aerodyne aerosol mass spectrometer (AMS) instruments, capable of on-line measuring chemical composition of non-refractory submicron aerosol species (Canagaratna et al., 2007; Ng et al., 2011b). These previous campaigns mostly focused on much polluted areas in eastern China, such as the Beijing–Tianjin–Hebei area (Takegawa et al., 2009; Huang et al.,2010; Sun et al., 2010, 2012, 2013, 2015; Zhang et al., 2011; Hu et al., 2013), the Yangtze River Delta (Huang et al., 2012, 2013), and the Pearl River Delta (He et al.,2011; Xiao et al., 2011; Lee et al., 2013),Xu et al. (2014) reported the chemical composition, and size distribution of submicron particulate matter (PM1)in Lanzhou in northwest China. In addition, Wang et al. (2016) recently used an Aerodyne soot particle-aerosol mass spectrometer (SP-AMS), for the first time in China, to investigate the occurrence of fullerene soot in ambient air. For summarization of AMS background site studies, as below: So far, several measurements and source analyses based on AMS have been conducted at background sites around the world. Sun et al. (2009) reported the composition and size distribution of NR-PM1at the Whistler Peak in Canada. Chen et al. (2009, 2015) conducted an AMS study to characterize submicron biogenic organic particles in the Amazon Basin. Ovadnevaite et al. (2011) demonstrated the occurrence of primary marine organic aerosol plumes on the west coast of Ireland. Du et al. (2015) described the aerosol composition using an ACSM at a national background monitoring station in the Tibetan Plateau in western China. These background site aerosol studies were all conducted in remote areas, which represent for global background atmosphere rather than regional background atmosphere, while regional background atmosphere is more critical to reflect the general picture of anthropogenic emissions in a hot polluted region. In this study, we performed online aerosol measurement field campaigns at two national air background sites in both northern and southern region in eastern China, which has a population of more than one billion and is characterized by worldwide high air pollution levels under high urbanization and industrialization.

2. As there already are so many AMS papers published in the past 15 years, it is difficult to see what is the significance and novelty of this paper. This should be made more clear, the novelty should not because you did AMS measurements at sites that are different from others, but instead you should state what scientific questions and what valuable findings you gained from your measurements that can advance our current understanding on aerosol chemistry? REPLY: Although many AMS measurements were conducted in the past years, this study also offers aerosol properties at two unique sites, which are regional background sites and critical for understanding the general picture of anthropogenic emissions in a hot polluted region, i.e., eastern China, where more than 90% of population in China live in. Based on the two background site campaigns, we found clearly different regional aerosol characteristics between South China and North China, e.g., aerosol compositions and major sources. Specially, our results suggested that possible sources influencing the two background sites may not only include emissions from the Chinese mainland but also include emissions from neighboring countries, which will no doubt improve the current scientific knowledge of regional-scale air pollution in East Asia. The above scientific judgment is more clearly stated in the revised text, such as in the introduction part and the conclusion part.

3. As the two measurements were conducted at different years, they have very different meteorological parameters, the discussion should consider and discuss more the meteorological effects, while current version is clearly lacking of such discussion. No meteorological data are shown. If so, the findings might be only for these two cases, having very limited scientific values for future and other studies. REPLY: The reviewer might ignore the meteorological parameters shown in Table 1. Actually, whether the two campaigns were conducted in the same year is not that important, because

the two sites have a distance of about 1900 km and thus cannot be influenced by the same weather system even in the same year. The difference of general air pollution characteristics between the two sites should be more determined by extensive regional emissions rather than by short-term local meteorology. In section 2.1, we give the information that no matter in 2013, 2014 or 2015, northern China all had a much higher PM2.5 concentration level than that in southern China, confirming that regional emissions are more important. In this study, we pay more attention to regional scale meteorology instead of local meteorology due to little local emission at the background sites. Therefore, back trajectories are more useful than local wind parameters to discuss meteorological effects in this study. We applied the TPSCF model, a statistical method based on back trajectories, to identify potential major source areas, which are not case-dependent and certainly useful for other aerosol studies in East Asia.

4. Regarding the quantification of organic nitrates, do the authors consider influences of metal nitrates? The AMS can measure nitrate that associated with sodium etc., although it is difficult to measure metals. Previous studies also pointed out that metal nitrates can have higher NO/NO2 ratios, so your estimation method is incorrect without considering this point. REPLY: At SCB, the concentrations of measured metals are very close to 0 because the very small "open-closed" difference mass spectra for V and W modes of Na, K, Al, Cu, Zn, and Pb. So the interferences of metal nitrates can be excluded. We make a clarification of this point in the revised manuscript as below: "On the other hand, significant existence of metal nitrates at SCB could be excluded due to non-detectable amounts of metals in the mass spectra."in Section 2.4.

5. The Vk diagram is quite limited in describing the formation processes of ambient OA in the reviewer's viewpoint. As there are so many possibilities that can influences the O/C and H/C ratios of ambient OA. The variation of O/C and H/C may not reflect the evolution processes at all. It maybe useful for chamber studies but should be discussed with cautions for ambient data. REPLY: We agree the reviewer's point and have made a clarification in the revised text to highlight that many other factors can

influence the slope in V-K diagram for ambient data, the Van Krevelen diagram is still useful for constraining the reactions that are responsible for the aging of OA (Hayes et al., 2013). And we modify some descriptions in Section 3.4 as below: "Heald et al. (2010) proposed using the Van Krevelen diagram to illustrate how reactions involving addition of functional groups fall along straight lines for ambient aerosol. Actually, many other factors may also lead to a variety of slopes in the Van Krevelen diagram in the case of ambient field measurements, while the Van Krevelen diagram is still useful for constraining the reactions that are responsible for the aging of OA (Hayes et al., 2013)."

6. Why a 3-factor solution was chosen for NCB site? It seems like 2-factor solution is also fine. In the supplement, it seems like the 2-factor solution is similar as the SCB site. Then why for NCB you chose 3-factor solution? Also, the Mass spectra of OOA1 and OOA2 in your 3-factor solution are quite similar, their diurnal patterns are similar too. Even if the authors insist to keep a POA factor for NCB site, this reviewer thinks OOA1 and OOA2 can be combined as one OOA factor. REPLY: There are two reasons why choosing the 3-factor solution for NCB as a better choice. Firstly, if we chose the 2-factor solution, the one with lower O:C ratio of 0.39 should be regarded as SV-OOA because it has a higher f44 of 6.5%, and thus we will miss HOA. However, high BC concentrations were observed at NCB, implying an HOA component should exist at NCB. Actually, in the quick review reports of this manuscript, it was suggested to split HOA from OOA at NCB by other reviewer. Secondly, although the mass spectra of OOA1 and OOA2 are similar, their time series are totally different, and their source areas are also different according to the TPSCF model analysis, indicating they are factors with different origins despite of similar O:C values.

7. A few technical comments (there maybe more, please check the MS carefully): page 2, line 15: "Valuable insights into the composition, sources, and evolution processes of was mostly found to be a submicron in China were obtained by the powerful on-line tools". What meaning? Please re-write this sentence. Page 5, line 7: a flow rate of 80 l min-1? should be 80 ml min-1. REPLY: All corrected.

---

## Author Comment (AC2) · 25 Jun 2016

1. It is difficult to compare the two studies directly since they were conducted in different years and different elevations. The authors should state clearly in the abstract that the two measurements were conducted in different years (2011 vs. 2015) and different elevations (21 m vs. 958 m) to avoid confusing the readers. REPLY: Suggestion taken. The information of sampling years and altitudes are added into the abstract.

2. Calling "Lake Hongze" as a site in northern China is not accurate, in fact, "Central eastern China" might be better. REPLY: Actually the northern China and southern China are just relative locations. Furthermore, many air mass back-trajectories for NCB shown in figure 1 were from the northern areas in China. As we mainly focus on

the regional pollutants transportation influence in this study, calling "Lake Hongze" as a northern China background site is reasonable.

3. Abstract, the authors claimed "the most aged OA in real ambient air ever reported in the literature", which is not correct. Please refer to Chen et al. (Geophys. Res. Lett., 42, 4182–4189, 10.1002/2015GL063693, 2015). REPLY: This sentence is deleted in the abstract and conclusion. And we modify some descriptions in Section 3.4 as below: "Note that the organic aerosol observed at SCB is very highly oxygenated compared to the ambient data ever reported in the literature, consistent with the previous finding that the atmospheric oxidizing capacity in southern China is unexpectedly high (Hofzumahaus et al., 2009)."

4. Page 2, line 14, rewrite this sentence. REPLY: Sentence is rewritten as "valuable insights on the composition, sources, and evolution processes of submicron particles in China were obtained through a dozen of field campaigns using various types of some powerful online tool."

5. Page 2, line 29, I didn't quite understand why the regional background air pollution is a critical factor in determining urban air quality. REPLY: Previous urban aerosol studies indicated that urban air pollution is not only just from the local emissions, but also from the regional pollutant transportation. Urban air quality significantly depends on the air pollutant concentrations input into the city, which highlights the importance of investigating regional background air pollution.

6. Page 3, line 1, "other instruments" actually refers to "AE31", I didn't see other collocated instruments. REPLY: Suggestion is taken. We use "an aethalometer (AE-31)" to substitute "other instruments ".

7. All the names of submicron aerosol species should be synchronized. For example, "Cl‐" vs Chl, "Organic in Figure 2" vs. "Organic aerosol", etc. REPLY: All corrected.

8. Page 5, line 7, flow is not correct. REPLY: Corrected.

9. Page 8, line 10, could you give a number for the overestimation? REPLY: This information has been added into Section 3.1 as below: "This overestimation could be less than 20% according to the ambient BC size distributions measured at an urban site in South China (Lan et al., 2011)."

10. Page 8, line 11,"Figure 2c"should be Figure 2e? REPLY: Corrected.

11. Figure 3a, the vertical lines did not match the maximum sizes. REPLY: The vertical lines in Figure 3a and 3b represent the mass median diameters rather than the size peak diameters, as stated in section 3.1.

12. Why the authors use different names for OOA components at the two sites, for example, OOA1 and OOA2 at Lake Hongze and SV-OOA and LV-OOA at Wuzhishan? REPLY: This is because of the specific solutions at the two sites. According to the previous literatures (Jimenez et al., 2009; Ng et al., 2010), two types of OOAs with different O/C ratios and volatilities have been observed in many ambient datasets: the OOA with higher O/C, which is more oxidized and aged, is referred to as low-volatility OOA (LV-OOA); the OOA with lower O/C, which is less oxidized and fresher, is referred to as semi-volatile OOA (SV-OOA). This type of solution applied to organic aerosol at SCB. However, the two OOA components at NCB had similar O/C ratios but quite different time series, and thus we named them OOA1 and OOA2 to avoid meaning that they have difference in terms of oxidation states. The splitting of organic aerosol into two components with similar O/C ratios were also observed in the 2008 Beijing Olympic Games AMS dataset (Huang et al., 2010).

13. Figures 3a, add figure legend for aerosol species. REPLY: Suggestion is taken.

14. Page 3, line 8 and line 10, same latitude and longitude for the two sites? REPLY: Mistake is corrected.

---

## Author Response (AR3)

1. I agree with the authors that the study of regional background sites is important to put e.g. urban measurements into perspective. One may indicate that this is important to do even at more sites and covering all seasons. In Europe, a comprehensive study was done mostly at regional background sites in different seasons using aerosol mass spectrometric measurements (Crippa et al., 2014). In the future, long-term measurements using aerosol chemical speciation monitors would be very helpful to cover all seasons at multiple sites. I suggest you cite the study of Crippa et al., 2014 and the need of more extended measurements at these and additional background sites in China.

**REPLY:**

We added the relevant descriptions in Section 3.6 as below,

"In summary, it is seen that the potential pollution source areas of the two background sites in China are on a regional scale rather than on a local scale, consistent with their nature as background sites. This highlights the need of long-term AMS measurements for all seasons at multiple sites in East Asia in order to have a comprehensive understanding of aerosol sources and transport in this region. Crippa et al. (2014) have recently well shown the advantages of multiple-site AMS measurements in acquiring insights of organic aerosol sources in Europe. "

2. The detection limit to measure more refractory elements by the open-closed difference is not very good. So I would not say that interferences of metal nitrates can be excluded. One may however say that one assumes and expects low contributions of metal nitrate in the submicron range (which would provide similar responses as organic nitrate) in contrast to the coarse mode where e.g. NaNO3 can be very abundant (you may find a reference for this).

**REPLY:**

We modify some descriptions in section 2.4 as below:

"In contrast to the abundant existence of metal nitrates in coarse mode particles (Huang et al., 2006), e.g., NaNO3, one may assume low contributions of metal nitrates in AMS detection in the submicron range."

Kouyoumdjian, H., and Saliba , N. A.: Mass concentration and ion composition of coarse and fine particles in an urban area in Beirut: effect of calcium carbonate on the absorption of nitric and sulfuric acids and the depletion of chloride, Atmos. Chem. Phys., 6, 1865-1877, doi:10.5194/acp-6-1865-2006, 2006.

3. Also in Hayes et al. (2013), they describe different results in the Los Angeles Basin that are due to mixing of some primary and secondary components. So I would tone it down even a bit more by saying .. "while the Van Krevelen diagram may still be useful for constraining..." above I would mention explicitly the mixing of different air masses and components/sources as possible reasons for variations in the slope.

**REPLY:**

We rephrased the sentences in section 3.4:

"Although many other factors, such as the mixing of different air masses and components/sources, may also lead to a variety of slopes in the Van Krevelen diagram in the case of ambient field measurements, the Van Krevelen diagram may still be useful for constraining reactions that are responsible for the aging of OA (Hayes et al., 2013)."

4. Change summarization to summary throughout the text.

**REPLY:**

We have corrected in the revised manuscript.

5. Page1: space between PM1 and components

**REPLY:**

Corrected

6. Page 2: before the year of the citations, sometimes spaces are missing. Please check throughout the manuscript.

**REPLY:**

Corrected.

7. Figure 3: make sure that size of Figures a, b are exactly the same including the text. add more space for the x and y-axes text. dM/dlogd .. nomenclature different in a, b versus c-h please make it the same. ug/m-3 is very hard to read on my printout.

**REPLY:**

Corrected.

8. Figure 4: check y-axis and x-axis text .. do them the same for all figures with the right distance to the axes numbers.

**REPLY:**

Corrected.

9. Figures 6c, 6d: add the x-axes at zero to the upper graphs (ticks and axes text not necessary)

**REPLY:**

Corrected.

[revised manuscript text omitted]